# Nature experiences affect the aesthetic reception of art: The case of paintings depicting aquatic animals

**Anne-Sophie Tribot[1], Daniel Faget[1], Thomas Changeux[2]***

**1** UMR TELEMMe, MMSH, Aix-Marseille University, CNRS, Aix-en-Provence, France, **2** Aix Marseille Univ, Université de Toulon, CNRS, IRD, MIO, Campus Luminy - OCEANOMED Bâtiment Méditerranée, Marseille, France

\* thomas.changeux@ird.fr

**Data Availability Statement:** The data underlying the results presented in the study are available from Figshare repository: https://doi.org/10.6084/m9.figshare.23592327.v1.

## Abstract

Art is a promising pathway to raise emotional engagement with nature, while enabling an indirect exposure to nature through aesthetic experience. However, the precise relationships between aesthetic experiences of art and experiences of nature remain unclear. The aim of this observational study is to highlight the effect of nature experiences on the aesthetic reception art, based on Early Modern paintings (16th-18th century). By focusing on marine ecosystems, that are difficult to directly interact with, the results presented are intended to explore whether marine activities and fish consumption affect the aesthetic reception of artworks depicting marine biodiversity. A photo-questionnaire survey based on four paintings has been conducted with 332 French participants with a diverse range of marine practices, fish consumption and artistic sensitivity. Fish consumption and value attributed to fish as food had a significant positive impact on the aesthetic reception, suggesting that taste and food consumption could be considered as a relevant nature aesthetic experience that elicits affective and emotional responses. Results also showed an indirect effect of fishing and diving on the aesthetic reception of paintings whose iconography relates with the observers' experiences. These findings are of particular interest in both environmental psychology and ecological mediation through art. This study brings evidences of the connection between art and nature experiences, and that art could be an innovative way of experiencing nature. Finally, this study also highlights the need to broaden the scope of nature experiences, for instance by including food.

## 1. Introduction

The aesthetic perception of nature is an essential part of the relationship between humans and nature, in the same way that art is a constituent element of human culture [1]. Within the many ways to experience nature, the engagement with nature's beauty is particularly important regarding current environmental challenges [2,3], since emotional and sensory engagement with nature seems to have more effect on pro-environmental behaviors than do

**Funding:** This work was supported by the Fondation de France, the Institut Méditerranéen pour la Transition Environnementale (ITEM), Excellence Initiative of Aix-Marseille University - A*MIDEX, a French "Investissements d'Avenir" programme, and Agence de l'eau Rhône Méditerranée Corse. The funding mainly paid AST's salary, ans also the small costs of DF, AMU agent and TC,IRD agent. The funders had no role in study design, data collection and analysis, decision to publish, or preparation of the manuscript.

**Competing interests:** The authors declare that no competing interests exist.

knowledge and theoretical education [4–7]. Aesthetic engagement with nature is thus a promising tool to improve motivation for conservation through nature experiences [8] while contributing to mental well-being [2]. In this perspective, environmental education has largely been dedicated to the mediation of biodiversity conservation through exposure to nature that enhance aesthetic experiences and promote engagement with nature's beauty [9]. Although promising, this research questions the way of applying these methods to inaccessible or inconspicuous nature. This is notably the case for aquatic ecosystems, which represent a very pertinent concern in the context of emotional engagement through nature aesthetic experience. They combine strong human and ecological issues, and as such are included in the 17 Sustainable Development Goals, while the relationships that humans develop with these ecosystems are mostly derived from indirect exposure (typically through food or cultural objects such as movies or pictures). Indeed, these ecosystems often remain inaccessible due to remoteness, mobility issues, lack of specific skills such as swimming, or to financial constraints [10]. This results in many beliefs and misapprehension in the perception of these ecosystems, that is often disconnected from ecological realities [11,12]. The resulting separation between human perception and ecological goals can act as a barrier to acceptance of conservation or restoration programs, and reveals the need to explore experiences that trigger significant affective and emotional responses.

Art is a worthy pathway in this context, because it allows to transcend the cognitive dimension of environmental concerns and to influences people's worldviews and life goals [13], while either enabling an aesthetic experience through indirect exposure to nature [14] or engaging with nature's beauty through artistic creation inspired by nature-based sources [15]. Given that art aesthetic experience here aims to substitute for the direct exposure to nature, there is therefore a need to define the factors though which they would mutually structure each other. More precisely, it is necessary to understand which experiences of nature have an effect on which dimensions of the aesthetic response. Surprisingly the field of psychology dedicated to the aesthetic sciences seems to have generally focused on art rather than nature, since works of art commonly elicit beauty judgments in people [16]. However, it is counterproductive to separate, on the one hand, nature experience and on the other hand art experience in this context, since aesthetic experience occurs in any situation that involves evaluative appraisal of objects [16,17]. This study therefore aims to reconcile art reception and nature experience, by considering the aesthetic reception of visual artworks representing the living world as an aesthetic experience of nature.

It has already been well demonstrated that artistic sensitivity and art knowledge influence the aesthetic reception of art [18]. In the same way, our hypothesis is therefore that experiences of nature may influence the aesthetic reception of artworks representing nature, in an iconographic context that echoes to the experiences of the observer. The observation of such an effect would constitute a new evidence of the link between nature experience and art aesthetic experience. This is of particular interest in environmental mediation through art, in order, for example, to take into account the previous nature experience of the target audience, and therefore the expected responsiveness [19], to adapt the artistic media used. Among the various ways of experiencing aquatic nature, food occupies an important place [20], as it represents the main regular contact with marine and freshwater fauna. It is thus relevant to question the perception of aquatic fauna as food in artistic and environmental mediation contexts, especially since overfishing represents one of the main concerns regarding aquatic biodiversity conservation [21]. Moreover, there is a clear link between aesthetics and food, since cooking could be interpreted as a kind of art, and taste and food consumption as aesthetic experiences [22]. Unlike numerous research that seeks to underlie the determinants of fish and seafood

consumption [23,24], this article conversely considers consumption as a determinant of aesthetic valuation.

Early modern (16th– 18th centuries) paintings represent a convenient way to integrate this dimension, since in European still-lifes of this period, aquatic fauna -and in particular fish- is mainly represented as food. The fish are generally represented in different types of scenes such as fishmonger's stalls, fishing products, kitchens, meals, or rather naturalistic paintings with a natural setting. This iconography is therefore particularly suited to the subject of this study. Other advantages of using Early Modern paintings relate to the temporal dimension of the relationship of humans with aquatic animals, and the changes of human perceptions of nature over time. This point is particularly important, since most ArtScience initiatives generally focus on post-19th c. art [25,26], while older art depicting nature in multiple aspects also deserves attention [14]. Finally, artworks from this period are likely to arouse multiple feelings ranging from curiosity, wonder, or appetite, to pity, disgust or unease [27], and thus elicit a multitude of positive or negative aesthetic responses.

In this context, this study aims to examine the effect of nature experience on the aesthetic reception of Early Modern paintings depicting aquatic biodiversity. This study thus proposes a step forward both in the ArtSciences field dedicated to environmental mediation and conservation biology, using an innovative and little explored material that is ancient art.

## 2. Material and methods

### 2.1 Participants

The survey was conducted with 332 online participants in France, aged between 16 and 95 years old. Participants were recruited using email distribution from local institutional mailing lists and advertisements. Women represented 60% of the participants, 6% were under 20, 46% were between 20 and 40, 36% were between 40 and 60, and 12% were over 60 years old. A large proportion of participants (54%) had higher managerial, administrative and professional occupations, and a job or a professional project related to the environment (70%). Fishing and spearfishing were regularly practiced by 13% of respondents, and scuba diving by 42%. Regarding artistic sensitivity, 20% of respondents had a job or professional project related to the visual arts, and 30% had visited a museum or exhibition at least 5 times in the past year.

### 2.2 Materials

Two sets of two paintings were randomly submitted in an online photo-questionnaire, for a total of four evaluated paintings (Fig 1). In order to limit the duration of the questionnaire and thus prevent the task from being too repetitive—and leading to biased responses -, the choice was made to assign only two paintings per participant instead of four. During each participation, one of the two sets of paintings was thus randomly assigned to each viewer. The number of ratings for each painting was 166 for 332 participants. The four paintings were Mediterranean still-lifes from the early modern period (16th-18th century) representing aquatic fauna. Stimuli have been chosen to show different types of scenes and iconographies, in which the marine fauna is represented in different contexts of representation (Fig 1): the diversity and abundance of resources, food and cooking, the fish market and pleasure, and the living animals. These paintings were chosen from a database of more than 300 painted works, by applying the following selection criteria: artworks from the same region (Mediterranean) to guarantee a homogeneity of artistic style, images available in high definition and under the public domain or CC0-License, or whose permission to publish has been obtained. The paintings should also be able to trigger positive or negative feelings depending on the viewers. Among the dozen works of art corresponding to these criteria, we selected the four that

**P1**

**P2**

**P3**

**P4**

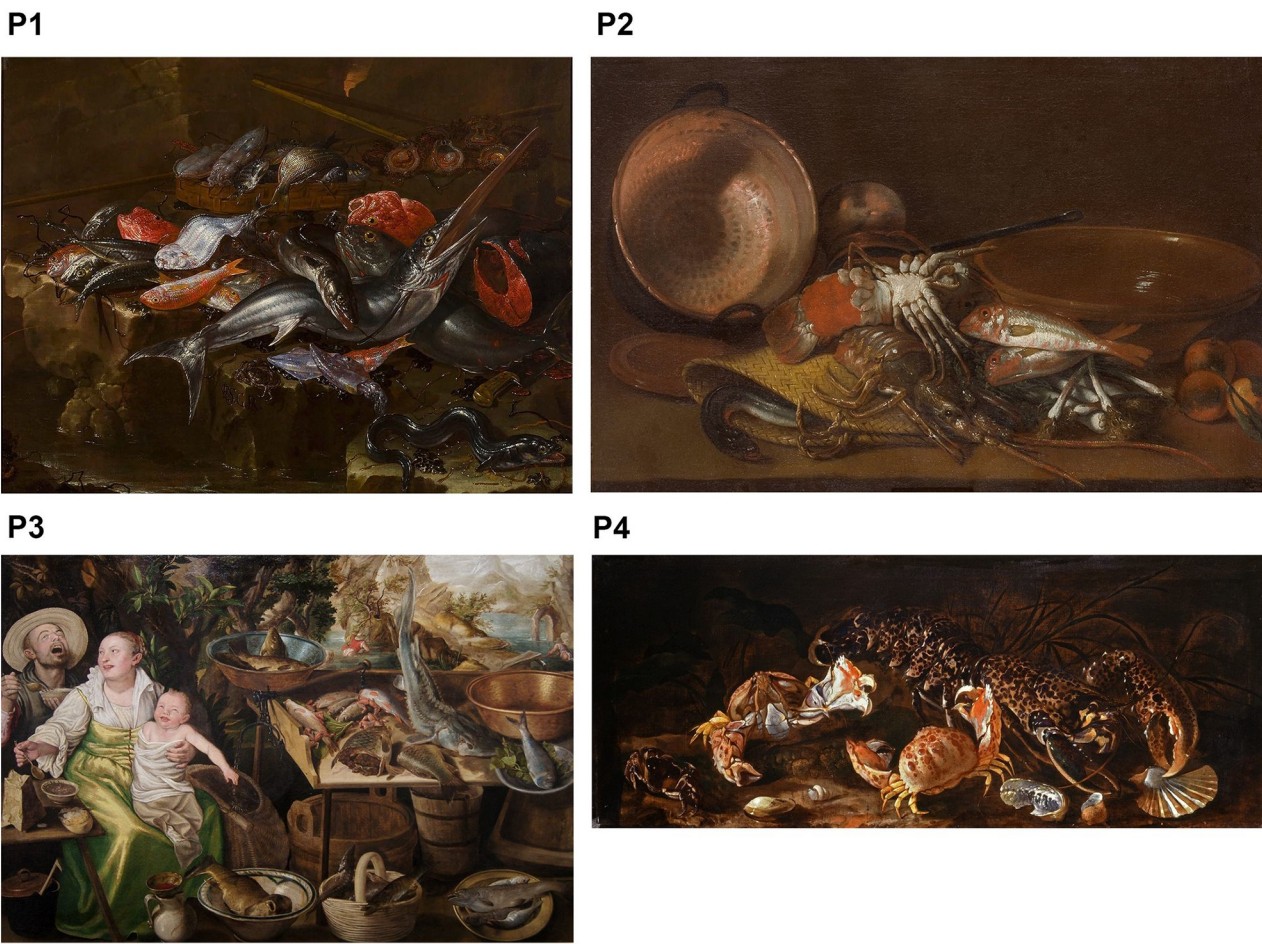

**Fig 1. Selected paintings.** P1: Giuseppe Recco—Naples (Italy) 1634—Alicante (Spain) 1695, *Pisces*, 1683. Private collection. Public domain CC0. Source: Commons.wikimedia.org. P2: Antonio Viladomat—Barcelona (Spain) 1678—Barcelona (Spain) 1755, *Still Life with Shellfish, Fish and Vessels*, 1710–1740. Prado National Museum. Public domain CC0. Source: Commons.wikimedia.org. P3: Vincenzo Campi—Cremona (Italy) 1536—Cremona (Italy) 1591, *The fishmongers*, 1579. Museum of La Roche-sur-Yon. Public domain CC0. Source: Commons.wikimedia.org. P4: Paolo Porpora—Naples (Italy) 1617—Rome (Italy) 1673, *Still life of fish and crustaceans*. Public domain CC0. Source: Commons.wikimedia.org.

seemed most representative of the different contexts of representation, based on free exploratory interviews.

In order to verify that these various representations of aquatic fauna were indeed perceived as belonging to different contexts, we collected written statements of what each painting evoked to observers. The results of the textual analysis of these statements are shown in S1 File.

Painting 1 (P1, Recco, *Pisces*, 1683) can be interpreted as a fish stall or fish products representing the richness of the sea. Through an ordered but complex composition, the purpose of this artwork is to show the diversity and the abundance of aquatic fauna. Overall dark, the painting is characterized by textures and light effects, with contrasting colors that highlight orange-red species, and with attention paid to the details while rendering reality. By showing a large number of dead fish out of the water, this painting also evokes in contemporary viewers the negative impact of human activities on marine biodiversity.

Painting 2 (P2, Viladomat, *Still Life with Shellfish, Fish and Vessels*, 1710–1740) can be interpreted as a kitchen scene. Kitchen utensils, fruits and vegetables evoke a recipe. Here aquatic

animals are mainly interpreted as food or gastronomic objects. The composition is simple but very homogeneous, with a marked degree of realism in a monochrome of ocher, orange and brown tones.

Painting 3 (P3, Campi, *The fishmongers*, 1579) is a market scene with a popular and comic dimension. The fish is present but it is not the main subject, which are the three characters: a couple and their child having their meal. Here, people are portrayed in their everyday life, and for this reason it can be described as genre scene painting. This painting evokes the sensations of pleasure and taste, family everyday life, diversity and abundance. Like P1, the fish stall, where fish that are visibly dead or prepared for cooking also—evoke for contemporary viewers the overconsumption of aquatic resources by humans.

Painting 4 (P4, Porpora, *Still life of fish and crustaceans*, 1617–1673) approaches naturalistic painting, with amazing living animals seen as in an aquarium, mainly crustaceans in movement with strange shapes and colors. It is an intimate scene that looks like a realistic cabinet of curiosities, and invites wonder and the discovery of the underwater world.

The objective of this survey is thus to identify whether the aesthetic reception of these artworks presenting different scenes with aquatic fauna vary according to the viewer's relationship to marine environments. This relationship was delimited through different types of exposure (i.e. experiences) to marine environment: marine activities as direct exposure (scuba-diving, snorkeling, fishing and spearfishing); and fish consumption as indirect exposure through the senses and the pleasure of taste. Since artistic sensitivity of participants is also likely to directly influence aesthetic reception, we also assessed the frequency of museum visits and identified the participants with a professional activity related to the visual arts (Fig 2). Finally, we also checked the effect of socio-demographic variables on aesthetic reception, such as age and socio-professional category.

## 2.3 Procedure

**2.3.1 Photo-questionnaire.** This study was declared to the ethics committee of Aix-Marseille University. Ethical approval was not required for this photo-questionnaire, conducted in accordance with the French public agency Commission Nationale de l'Informatique et des Libertés (CNIL): it was strictly anonymous, no personal data was collected, the participants were informed of the general purpose of the study and the processing of the data, the contact details of the researchers were provided to the participants, participants gave their consent by checking a box at the beginning of the questionnaire. The study was carried out from September

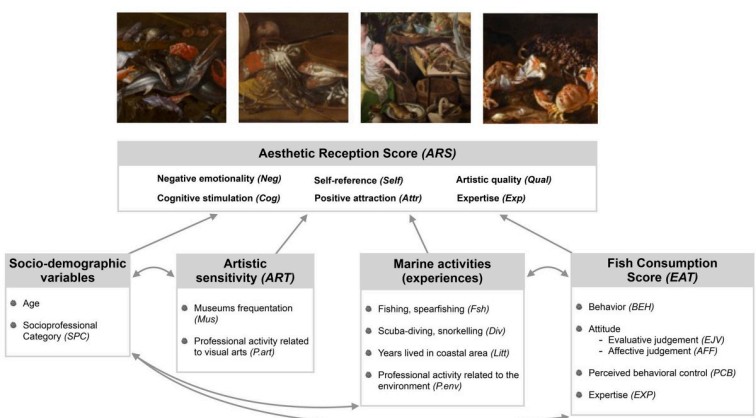

**Fig 2. Summary of the measured variables.** Arrows represent the tested relationships.

2022 to November 2022, on an adult audience (over 18 years old), recruited by an email information campaign among Aix-Marseille University staff and national academic mailing lists.

The photo-questionnaire consisted of several steps (see questionnaire provided in S2 File): (i) *Instructions*: the viewers were informed of the different steps of the questionnaire, they were asked to be in a quiet place, at a computer, and if possible, with a large screen. There was no time limitation. (ii) *Observation*: two photos of the paintings from the random set were sequentially shown in high definition on the screen. The viewer could observe the painting as long as he/she wished and could return to it at any time. (iii) *Art Reception Survey*: the viewer rated two paintings using a questionnaire. A textual content field was also available to viewers, to express in a few words what the paintings evoked for them. (iv) *Fish consumption survey*: viewers provided information on their fish consumption habits and the values they attribute to fish as food, using a questionnaire. (v) *Marine practices and experiences*: additional questions were intended to identify persons engaged in fishing or spearfishing, and/or scuba diving. We also asked for the number of years lived in coastal areas, and identified participants with a professional activity related to the environment (vi) *Artistic interest*: we estimated artistic interest by asking how many times the participants had visited a temporary exhibition or a museum in the last twelve months, and by identifying participants with a professional activity related to visual arts (vii) *Socio-demographic variables*: we asked participants their age, gender and socio-professional category (SPC).

**2.3.2 Assessment of aesthetic reception.**   To assess the aesthetic experience of the viewers, we used the Art Reception Survey adapted from Hager at al. [1]. This self-assessment scale aims to precisely measure the different structuring factors of the aesthetic experience. Based on the model of aesthetic appreciation and aesthetic judgement proposed by Leder et al. [28], this scale measures 6 factors: Cognitive stimulation, Negative emotionality, Expertise, Self-reference, Artistic quality and Positive attraction (see S3 and S4 Files). We used a simplified version of the scale, by selecting the most informative items tested by Hager et al. (items with Loading > 0.7). Within the factor 'Negative emotionality' we added an item dedicated to the feeling of anger, as we hypothesized that some paintings may trigger such feelings due to the representation of fish out of water (Particularly for P1 and P2). For each item, the participants were asked to rate their endorsement on a five-point rating scale, ranging from 1 = completely disagree, 2 = rather disagree, 3 = neither nor, 4 = rather agree, 5 = completely agree. We determined the aesthetic reception score (ARS) of participants by calculating the mean of each factor for each painting. Factor scores were calculated by averaging each corresponding item. We also calculated the overall ARS for each painting by averaging the total scores obtained for each factor.

**2.3.3 Viewers' factors.**   *Fish consumption*. In order to assess whether fish consumption behavior and the value attributed to fish as food influence aesthetic reception, we used a self-assessment scale adapted from Verbeke and Vackier [29], based on the Theory of Planned Behavior [30] that links attitude, subjective norms and perceived behavioral control to behavior. In order to shorten the duration of the questionnaire, we have chosen to use a simplified scale by selecting the most informative items according to Verbeke and Vackier [29]. The scale is thus composed of 13 items classified among 5 factors: Behavior; Attitude (evaluative and affective judgements); Perceived behavioral control; and Past experiences (see S3 and S4 Files). For each item, the participants were asked to rate their endorsement on a five-point rating scale, ranging from 1 = completely disagree, 2 = rather disagree, 3 = neither nor, 4 = rather agree, 5 = completely agree. The final fish consumption score (EAT) for each participant was calculated by averaging each factor. Factor scores were calculated by averaging each corresponding item.

We expected that participants who regularly consume fish, and who have a more positive attitude and experience in preparing and eating fish are more likely to have a higher aesthetic reception. Given that fish as depicted in Early Modern still-lifes is mostly represented as food (particularly in P1, P2 and P3), we expect that fish consumption score mainly affects the factor 'Self-reference' of aesthetic reception.

*Marine activities*. We identified participants that practice fishing or spearfishing (Fsh), and/ or scuba diving and snorkeling (Div). We also asked for the number of years lived in coastal areas (Litt) and identified participants who have a professional activity related to the environment (P.env). The aim is to test the hypothesis that viewers who are more often directly exposed to marine life are more likely to have strong aesthetic experiences in front of paintings representing the aquatic biodiversity (particularly for P4).

*Artistic sensitivity*. Since regular exposure to art can influence aesthetic reception [18], we asked participants how many times they had visited a temporary exhibition or a museum in the last twelve months (Mus). We hypothesized that participants who regularly visit museums are more likely to obtain a higher aesthetic reception score, regardless of the painting. We also identified participants with a professional activity related to visual arts (P.art).

*Socio-demographic variables*. We also tested the effect of variables such as age and socio-professional category, since they are likely to influence fish consumption, marine practices and the museum attendance.

## 2.4 Statistical analyses

**2.4.1 Aesthetic reception scores of paintings.** We computed the aesthetic reception score (ARS) of each painting by calculating the mean and the standard deviation of each of the six factors (Cognitive stimulation, Negative emotionality, Expertise, Self-reference, Artistic quality and Positive attraction). In order to assess the validity of the observed ARS score of paintings, we first computed an exploratory factor analysis (see Supporting information). Overall, all items had a Cronbach's alpha value higher to 0.7, but a few had item-rest correlation values less than 0.2 ('This painting is pleasant', 'This painting disgust me', 'This painting makes me feel afraid') and a uniqueness value close to 0.7 ('This painting is very innovative', 'I know this painting' and 'I can relate this painting to its art historical context'). Given that the construction of the six factors based on the items' contributions did not perfectly match that of Hager et al. [1], we have therefore chosen to keep the original ARS factors from the authors, and perform a confirmatory factor analysis based on this model (see S4 File). The parameters of the model obtained had a Comparative Fit Index (CFI) of 0.87, a Tucker Lewis Index (TLI) of 0.84, and a Root Mean Square Error (RMSE) of 0.08.

**2.4.2 Fish consumption scores of participants.** We computed the fish consumption score (EAT) of each participant by calculating the mean and the standard deviation of each of the five measured factors (Behavior; Evaluative and Affective judgements; Perceived behavioral control; and Past experiences). In order to assess the validity of the observed EAT scores, we used an exploratory factor analysis followed by a confirmatory factor analysis (see S3 and S4 Files), based on the model from Verbeke and Vackier [29]. The parameters of the obtained model had a CFI of 0.98, a TLI of 0.96, and a RMSE of 0.05.

**2.4.3 Marine activities, Artistic sensitivity and Sociodemographic variables.** In order to explore the links between the different variables that are not based on a psychometric scale, we performed a Principal Component Analysis (PCA), with the aim of gathering the associated variables into new factors (see S5 File). The four resulting factors explained 64% of the variance, based on the grouping of the following variables: (i) socio-professional category and age (17%); (ii) fishing and number of years lived in a coastal area (FSH); (iii) diving and

professional activity related to the environment (DIV); and (iv) frequency of museum visits and professional activity related to the visual arts (ART). These four factors were then used in the final structural equation model as latent variables.

**2.4.4 Effect of viewers' factors on aesthetic reception.** In order to assess the correlations between the viewer's factors and each of the six ARS factors, we performed Pearson's correlation tests. We then carried out a structural equation model (SEM) in order to identify the directionality of the effect of viewer's factors on aesthetic reception. The latent variables used in the model were defined on the basis of the previous factorial analyses. The effects of each of these latent variables on ARS scores were then assessed using multiple regression. Additional regressions between significantly correlated intra-factorial variables were also added.

# 3. Results

## 3.1 Aesthetic reception of the paintings

Overall, the mean aesthetic reception scores of the four paintings were not significantly different. However, the results showed a noticeable difference between P3 which obtained the highest mean ARS score, and P2 which had the lowest mean ARS score (Table 1). Regarding the mean of each factor considered separately, P3 obtained the lowest score of Artistic quality and Cognitive stimulation. P1 obtained the highest score of negative emotionality, conversely P4 got the highest score for Attractivity.

## 3.2 Correlations between viewers' factors and aesthetic reception factors

Pearson's tests revealed significant correlations between viewer's factors and ARS factors (Table 2). Fish consumption score (EAT) was the one most often correlated with ARS factors, particularly with Attractiveness, Expertise, and Negative emotionality (negative correlation). Fishing (FSH) and diving (DIV) were also positively correlated with Self-reference aesthetic factor for all paintings except P3, suggesting that the iconographic content from which humans are absent directly echoes the past experiences of divers and fishers. Artistic sensitivity (ART) showed a high correlation with ARS of P3 and P4, especially for Expertise and Cognitive stimulation, while it was not correlated with ARS of P1 and P2. This suggest that scene genre and naturalist paintings arouse more aesthetic interest for art enthusiasts than still-lifes.

**Table 1. Mean and standard deviation of each Aesthetic reception score (ARS) factor, separated for each painting (P1 to P4).**

| Factor | P1 | | P2 | | P3 | | P4 | |
|---|---|---|---|---|---|---|---|---|
| | *M* | *SD* | *M* | *SD* | *M* | *SD* | *M* | *SD* |
| Qual | 2.383 | .711 | 1.837 | .758 | 2.442 | .640 | 2.275 | .764 |
| Attr | 1.738 | 1.013 | 2.030 | .997 | 2.051 | .925 | 2.762 | .889 |
| Neg | 1.483 | .896 | .816 | .755 | 1.248 | .860 | .582 | .641 |
| Exp | 1.042 | .854 | .839 | .785 | 1.100 | .859 | .847 | .879 |
| Self | 1.018 | 1.063 | .978 | 1.145 | .857 | .959 | .886 | 1.058 |
| Cog | 2.282 | .939 | 1.708 | .943 | 2.535 | .816 | 2.169 | .956 |
| ARS | 1.658 | .466 | 1.368 | .507 | 1.706 | .445 | 1.587 | .531 |

M = Mean; SD = Standard deviation. Qual = Artistic quality; Attr = Attractivity; Neg = Negative emotionality; Exp = Expertise; Self = Self-reference; Cog = Cognitive stimulation.

**Table 2. Correlations between the Aesthetic reception score (ARS) factors of each painting and the viewer's factors.**

| Factor | P1 | | | | Factor | P2 | | | |
|---|---|---|---|---|---|---|---|---|---|
| | ART | EAT | DIV | FSH | | ART | EAT | DIV | FSH |
| Qual | .07 (.346) | .08 (.315) | .07 (.398) | .08 (.314) | Qual | -.01 (.895) | .15 (.05) | .02 (.809) | .05 (.552) |
| Attr | .02 (.794) | **.22 (.005)** | .01 (.856) | .14 (.080) | Attr | .16 (.040) | **.22 (.005)** | -.04 (.586) | .12 (.124) |
| Neg | -.01 (.861) | **-.30 (.000)** | .00 (.988) | -.15 (.051) | Neg | -.04 (.642) | **-.32 (.000)** | -.01 (.887) | **-.24 (.002)** |
| Exp | .10 (.220) | **.31 (.000)** | .00 (.958) | .18 (.026) | Exp | .04 (.662) | **.25 (.001)** | -.08 (.334) | .17 (.026) |
| Self | -.05 (.534) | .11 (.180) | **.23 (.003)** | **.22 (.006)** | Self | -.03 (.749) | .17 (.034) | **.24 (.002)** | **.20 (.013)** |
| Cog | .19 (.013) | .13 (.113) | .04 (.659) | .09 (.261) | Cog | .04 (.597) | .17 (.030) | .07 (.371) | .15 (.057) |
| ARS | .10 (.211) | .18 (.023) | .12 (.131) | .18 (.018) | ARS | .05 (.498) | **.21 (.007)** | .08 (.320) | .16 (.049) |
| Factor | P3 | | | | Factor | P4 | | | |
| | ART | EAT | DIV | FSH | | ART | EAT | DIV | FSH |
| Qual | .15 (.061) | .11 (.180) | -.09 (.283) | .13 (.092) | Qual | .09 (.264) | **.21 (.006)** | .08 (.343) | .04 (.583) |
| Attr | .16 (.037) | **.21 (.007)** | -.11 (.178) | .11 (.183) | Attr | .18 (.019) | .14 (.066) | .08 (.287) | -.02 (.778) |
| Neg | **-.20 (.011)** | **-.23 (.004)** | .12 (.138) | -.07 (.376) | Neg | -.08 (.328) | -.10 (.227) | .02 (.829) | .02 (.851) |
| Exp | **.29 (.000)** | **.21 (.007)** | -.03 (.689) | .03 (.707) | Exp | **.27 (.001)** | **.22 (.006)** | .03 (.703) | -.01 (.910) |
| Self | .14 (.073) | **.23 (.003)** | .16 (.041) | .19 (.013) | Self | .16 (.047) | **.25 (.002)** | **.22 (.005)** | **.23 (.003)** |
| Cog | **.23 (.003)** | .18 (.025) | -.07 (.356) | .04 (.594) | Cog | **.20 (.011)** | **.20 (.012)** | .11 (.166) | .10 (.204) |
| ARS | **.24 (.002)** | **.23 (.003)** | .01 (.945) | .14 (.082) | ARS | **.24 (.002)** | **.27 (.000)** | .16 (.042) | .11 (.157) |

Values correspond to the Pearson correlation coefficient r. Values in brackets show the p-value of the Pearson correlation test. Bold values represent significant correlations. Qual = Artistic quality; Attr = Attractivity; Neg = Negative emotionality; Exp = Expertise; Self = Self-reference; Cog = Cognitive stimulation. ART = Artistic sensitivity (museum frequentation and professional activity related to visual arts); EAT = Fish consumption score; DIV = Diving (coupled with professional activity related to environment); FSH = Fishing (coupled with number of years lived in a coastal area).

### 3.3 Effect of viewers' factors on aesthetic reception

The SEM model (Fig 3) shown a direct significant effect of artistic sensitivity and fish consumption score on aesthetic reception, including an indirect effect of scuba-diving and fishing on fish consumption score. Note that professional activity related to the environment and number of years lived in a coastal area had a significant effect on diving and fishing, respectively. Age and SPC were included in the model as belonging to fish consumption (EAT) latent variable, because the model obtained better validity parameters with this classification. These two variables therefore had an indirect effect on aesthetic reception, since they positively influenced fish consumption.

## 4. Discussion

Engagement with nature's beauty is an important determinant of motivation for the conservation of biodiversity. Although this particular aspect of human-nature relations has mostly been explored through the prism of direct experiences of nature, exposures to nature through art deserve particular attention in the case of inaccessible environments such as aquatic ecosystems. However, the characterization of the link between art reception and nature experience needs to be better defined. As a first step, this observational study thus aims to explore the effect of previous nature experiences of the observer on the aesthetic reception of art depicting nature, based on Early Modern paintings. A photo-questionnaire survey based on four paintings has been conducted on 332 French participants with a diverse range of marine practices, fish consumption and artistic sensitivity.

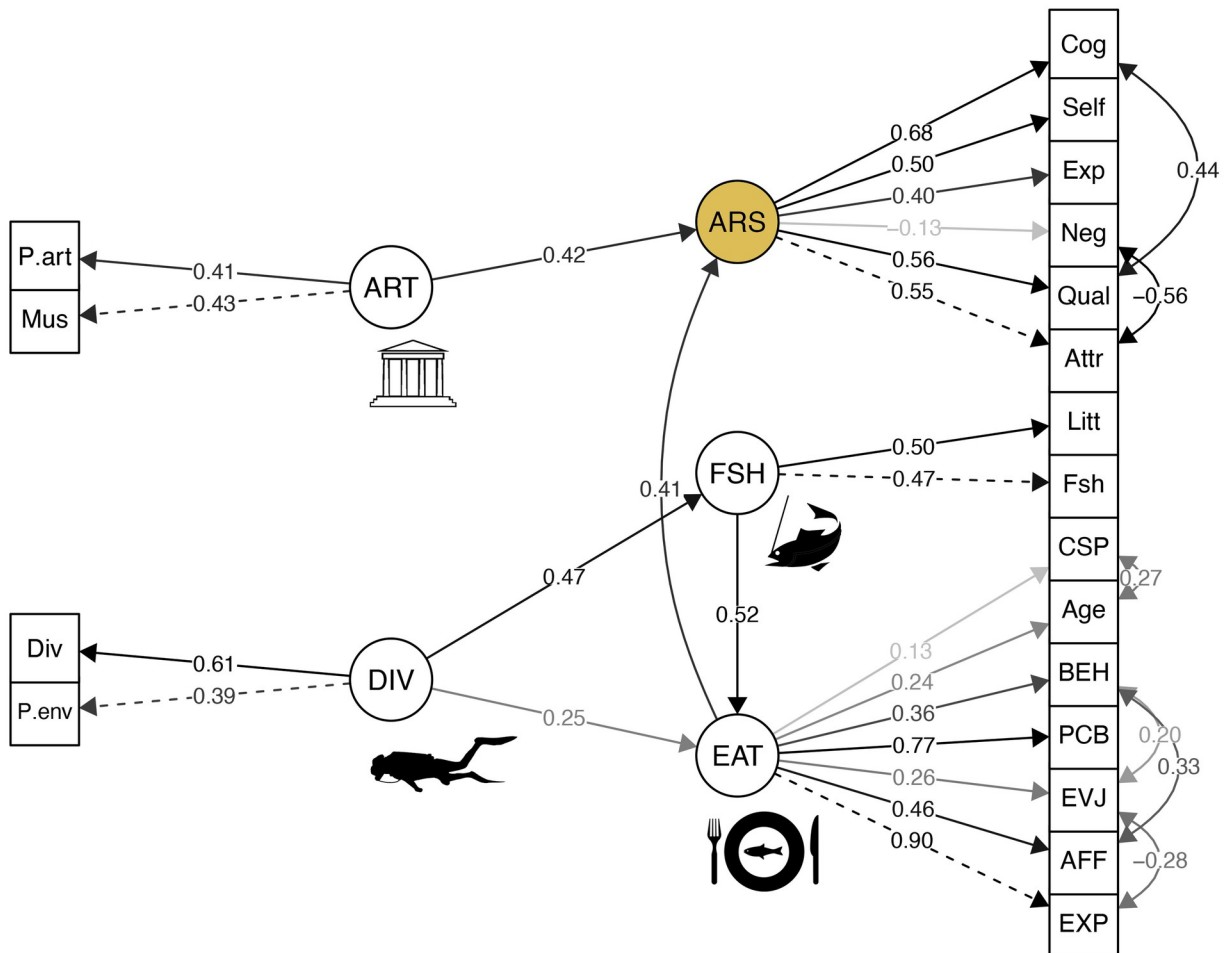

**Fig 3. Path diagram of the structural equation model (SEM) of the relationship between observer variables and global aesthetic reception.**
Simple arrows show direct relationships. Double-headed arrows show correlation between measured variables (only significant correlations are shown). The color of the arrows shows the sensitivity of the relationship, specified by the number on the line (standardized regression weights). For the variables corresponding to the abbreviations, see Fig 2. Fit indices: CFI = 0.821; TLI = 0.746; RMSEA = 0.074. Exploratory and Confirmatory Factor Analyses used to build the model are shown in S3 and S4 Files.

## 4.1 Fish consumption

One of the most important contribution of this study concerns fish consumption and valuation of fish as food: Aesthetic reception of the four paintings has been particularly sensitive to the fish consumption score that aggregates Behavior; Evaluative and Affective judgements; Perceived behavioral control; and Past experiences. More precisely, this specific kind of exposure to aquatic organisms increases the attractivity of the paintings, and decreases negative emotionality. Thus, it seems that observers who regularly consume and enjoy fish as food tend to judge the still-lifes to be more beautiful and pleasurable.

This result seems counter-intuitive given previous research on the assessment of consumed animals, that are typically denied any kind of positive evaluation: meat eaters tend to reduce mind attribution to animals, and ascribe to edible animals lesser mental capacities [31,32]. Particularly, Loughnan et al. [32] showed that aquatic animals (fish, lobster, prawn and crab) were ranked among the most edible animals with the lowest mental capacities. According to the authors, the dissonance related to their consumption therefore may be less evident than for

other animals such as mammals. One hypothesis regarding our results is that eating fish might be considered a more morally acceptable practice, and thus more conducive to a positive aesthetic response. However, it is important to note that scuba diving had a direct effect on fish consumption. This sporting and contemplative activity seems *a priori* incompatible with a negative valuation of aquatic species. It would thus suggest that the consumption of these animals does not depress the affective and aesthetic value attributed to them. Another hypothesis, which needs to be explored, is that people with a strong taste for fish could consider cooking and eating as an aesthetic experience in itself. In addition, the cultural dimension of this product probably plays a major role here. Given that the fish consumption was influenced by fishing and the number of years lived in the coastal area, fish could be considered as a gastronomic symbol of a region, and as a factor of cultural identity. In France, the most consuming regions are unsurprisingly the coastal areas, including marked regional specificities, according to a report published in 2021 by FranceAgriMer, *La consommation des produits aquatiques en 2020* [33].

It therefore seems important to question here the purely utilitarian vision of nature within the act of eating. Taste and food consumption should be considered as a relevant nature aesthetic experience that elicits affective and emotional responses, and reveal the existence of an intrinsic and relational heritage value [34]. In this sense, engaged consumption might not systematically be unfavorable to the motivation for the sustainable conservation of biodiversity, since these aesthetic experiences could trigger concerns about overfishing and consumption-related issues. Conversely, it is important to acknowledge that taste enjoyment related to fish as food may also arouse the desire to consume and therefore influence behaviors that not favorable to sustainable aquatic biodiversity, especially for species whose consumption results from overfishing or intensive aquaculture. Thus, we hypothesize that the act of eating could be considered as an aesthetic experience of nature that could generate pleasure and interest in marine life, capable of having a favorable effect on behavior, on the condition that this specific behavior is engaged -both physically and morally- and thoughtful. This perspective, however, needs to be tested experimentally.

## 4.2 Fishing and scuba-diving

Fishing and scuba-diving had an indirect effect on global aesthetic reception, by influencing fish consumption. Scuba-diving was influenced by professional activity related to the environment, and fishing was partly determined by the number of years spent in a coastal area. These marine practices were positively correlated to P1, P2 and P4 through Self-refence. It clearly appears here that experiences of marine environments act on the aesthetic reception of still-lifes through a mere exposure effect. In other words, these representations directly echo the experiences of observers, and create a sense of identification and familiarity with the represented species [19,35]. This hypothesis is strengthened by the fact that scuba-diving and fishing were not correlated to the aesthetic reception of P3, whose subject is not aquatic animals but human characters.

## 4.3 Age and SPC

Age and Socio-professional category had an indirect effect on global aesthetic reception, by influencing fish consumption. This result is congruent with statistics regarding French fish consumption, which increases with age and income level [33].

## 4.4 Artistic sensitivity

Unsurprisingly, artistic sensitivity had the highest impact on the global aesthetic reception of the paintings. This factor, which includes the frequency of museum visits and professional

activity related to visual arts, was particularly correlated to the aesthetic score of P3 and P4, especially for Expertise and Cognitive stimulation. It thus seems that observers with a strong artistic sensibility and knowledge had stronger aesthetic responses trough the cognitive dimension. P3 represent a classical market scene largely inspired by famous Flemish painters such as Joachim Beuckelear, and correspond to the genre scenes to which museum audiences in Europe are commonly exposed. Regarding P4, it is a naturalist painting in which the animals are represented alive and are integrated into the environment, which makes it particularly original for its time, likely to pique the interest of experienced observers.

### 4.5 Focus on iconographic interpretation

The textual statements related to the iconographic interpretation of the paintings (see S1 and S2 Files) revealed information that helps to understand their general aesthetic reception.

In particular, these interpretations are differentiated by positive and negative valences. Positive interpretations related to richness, abundance, diversity and wonder, and to a lesser extent to taste and appetite. Conversely, negative interpretations referred to overconsumption, overfishing and animal suffering, and were felt by people that were not engaged in marine activities and fish consumption. These perceptions are obviously anachronistic, since the artists of that period did not aim to alert the public to environmental issues. Although it is difficult to assess how negative emotions contribute to aesthetic pleasure [36], these interpretations demonstrate how artworks from the past can echoes with our contemporary environmental concerns. These paintings also seem to trigger a feeling of empathy in an audience with little exposure to marine ecosystems, and could therefore be considered as a possible mediation tool. Regarding P4, which shows living species from a rather naturalistic approach, the interpretations were much more consensual and its aesthetic reception was overall more positive.

Whether positive or negative, these aesthetic experiences arouse interest and convey emotional involvement with the animals depicted. Among the several possible aesthetic responses, the observation of these artworks therefore invites reflection and allows us to question the evolution of our perception of the environment and its resources over time. These artworks seem to allow us to connect with aquatic worlds, otherwise inaccessible, through the senses and emotions.

## 5. Conclusions

The results of these observational study highlight a link between perception of artistic representations of nature, and personal experiences with the living world. This link can be explained by exposure effect and consequently by a feeling of familiarity, and of cultural, affective and emotional attachment. In this way, taste and food consumption could be considered as a relevant nature aesthetic experience that elicits affective and emotional responses, and could be beneficial–or at least not deleterious–for biodiversity conservation. Aesthetic experience of art could thus be a relevant entry point to discuss the personal history and relationship to nature of the observers.

We acknowledge that this observational study has limitations. First, the sample was intended to measure the effect of marine practices on the aesthetic reception of still-lifes, and therefore constitutes a specific study case that it would be unwise to generalize. Furthermore, this study was conducted specifically in France, based on European artworks from a particular period. This study is therefore culturally and artistically biased. It would thus be necessary to test the aesthetic reception of these artworks by observers from other countries to evaluate more precisely the effect of cultural dimensions, such as place attachment [35]. In addition, this study focuses on fish consumption, yet the aquatic animals that are both commonly

consumed and depicted in still life paintings are not limited to fish. A similar study could be conducted by focusing on other groups, such as crustaceans or molluscs, whose aesthetic perception could be slightly different. Finally, this study relied on a photo-based method, and therefore cannot replace an experience carried out in-situ [37]. However, such an approach offers relevant perspectives intended to show artworks outside of museums, in order to reach an audience that does not visit them [38].

In conclusion, although it is impossible to assert that the aesthetic experience of art can be directly substituted for an aesthetic experience of nature—given its multi-sensory dimension —this study demonstrates however the fundamental connection between art and nature experiences. Art could therefore be an innovative way of experiencing nature. By providing further evidence that art should be considered a promising means of engaging with biodiversity by indirect exposure and through aesthetic experience, this study is of particular interest for aesthetic learning and biological conservation, and is a contribution to the development of ArtScience initiatives dedicated to the promotion of biodiversity through affective and emotional dimensions. This study also highlights the need to redefine and broaden the scope of nature experiences, for example by including food. This line of research would make it possible to add a new sensory dimension—in this case taste—to Human-Nature relationships.

This result constitutes a first step, the next one consists of exploring the reverse relationship: studying precisely how art can influence the perception of nature. This perspective is of particular interest for environmental psychology and ecological mediation, and to address the potential role of museum collections and ancient art as a means of engaged exposure to physically inaccessible ecosystems [14].

## Supporting information

**S1 File. Semantic analysis of the representations of the paintings according to the observers.**
(PDF)

**S2 File. Questionnaire.**
(PDF)

**S3 File. Exploratory factor analysis of aesthetic reception scale and fish consumption scale.**
(PDF)

**S4 File. Confirmatory factor analysis of aesthetic reception scale and fish consumption scale.**
(PDF)

**S5 File. Principal component analysis of marine activities, artistic sensitivity and socio-demographic variables.**
(PDF)

## Author Contributions

**Conceptualization:** Anne-Sophie Tribot.

**Data curation:** Anne-Sophie Tribot.

**Formal analysis:** Anne-Sophie Tribot.

**Funding acquisition:** Daniel Faget, Thomas Changeux.

**Investigation:** Anne-Sophie Tribot.

**Methodology:** Anne-Sophie Tribot.

**Project administration:** Thomas Changeux.

**Supervision:** Daniel Faget, Thomas Changeux.

**Validation:** Thomas Changeux.

**Writing – original draft:** Anne-Sophie Tribot, Thomas Changeux.

**Writing – review & editing:** Anne-Sophie Tribot, Thomas Changeux.

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
