## [Decision Letter · Decision Letter 0]

2 Nov 2023

PONE-D-23-28152Nature experiences affect the aesthetic reception of art: The case of paintings depicting aquatic biodiversityPLOS ONE

Dear Dr. Changeux,

Thank you for submitting your manuscript to PLOS ONE. After careful consideration, we feel that it has merit but does not fully meet PLOS ONE’s publication criteria as it currently stands. Therefore, we invite you to submit a revised version of the manuscript that addresses the points raised during the review process.

We look forward to receiving your revised manuscript.

Kind regards,

Dharmendra Kumar Meena

Academic Editor

PLOS ONE

Journal Requirements:

2. You indicated that ethical approval was not necessary for your study. We understand that the framework for ethical oversight requirements for studies of this type may differ depending on the setting and we would appreciate some further clarification regarding your research. Could you please provide further details on why your study is exempt from the need for approval and confirmation from your institutional review board or research ethics committee (e.g., in the form of a letter or email correspondence) that ethics review was not necessary for this study? Please include a copy of the correspondence as an ""Other"" file.

"NO"

4. Please ensure that you include a title page within your main document. You should list all authors and all affiliations as per our author instructions and clearly indicate the corresponding author.

6. We note that Figure 1 in your submission contain copyrighted images. All PLOS content is published under the Creative Commons Attribution License (CC BY 4.0), which means that the manuscript, images, and Supporting Information files will be freely available online, and any third party is permitted to access, download, copy, distribute, and use these materials in any way, even commercially, with proper attribution. For more information, see our copyright guidelines: http://journals.plos.org/plosone/s/licenses-and-copyright.

A.) You may seek permission from the original copyright holder of Figure 1 to publish the content specifically under the CC BY 4.0 license. 

B.) If you are unable to obtain permission from the original copyright holder to publish these figures under the CC BY 4.0 license or if the copyright holder’s requirements are incompatible with the CC BY 4.0 license, please either i) remove the figure or ii) supply a replacement figure that complies with the CC BY 4.0 license. Please check copyright information on all replacement figures and update the figure caption with source information. If applicable, please specify in the figure caption text when a figure is similar but not identical to the original image and is therefore for illustrative purposes only.

**Additional Editor Comments:**

Article can not be processed in its present form and recommend a through major revisions

Reviewers' comments:

Reviewer's Responses to Questions

**Comments to the Author**

1. Is the manuscript technically sound, and do the data support the conclusions?

Reviewer #1: Partly

Reviewer #2: Yes

2. Has the statistical analysis been performed appropriately and rigorously? 

Reviewer #1: Yes

Reviewer #2: Yes

3. Have the authors made all data underlying the findings in their manuscript fully available?

Reviewer #1: Yes

Reviewer #2: Yes

4. Is the manuscript presented in an intelligible fashion and written in standard English?

Reviewer #1: Yes

Reviewer #2: No

5. Review Comments to the Author

Reviewer #1: This manuscript explores the relationship between aesthetic experiences of art and experiences of nature, focusing on aquatic animals. The idea and perspective of this manuscript are intriguing, but some contents are not always clear. The authors need to clarify the following:

Specific comments:

1. Line 204-220. The use of Marks (1)-(7) for labeling may be confused with reference citation numbers. The way they are used in Line 307 and 308 appears to be clearer.

2. I suggest the author includes their questionnaire as supplementary information. This would greatly assist readers in quickly understanding their work.

3. Based on their questionnaire, the authors concluded that “fish consumption and value attributed to fish as food had a significant positive impact one the aesthetic reception (Line 23-24, and also Line 293-293)”. Then they proposed that “taste and food consumption could be considered as a relevant nature aesthetic experience that elicits affective and emotional responses (Line 24-25)”. This conclusion and discussion seem reasonable.

However, on this basis, the authors emphasized that “taste and food consumption could be beneficial for biodiversity conservation (Line 26, and also Line 416-419)”. I found the logic behind this notion in the manuscript unclear. While it is true that these experiences would increase the attractivity of the paintings and decreases negative emotionality, it seems challenging to predict whether this would ultimately be beneficial or detrimental to biodiversity conservation. Someone might be more concerned about the sustainable conservation of biodiversity. Conversely, it may also arouse people's desires, potentially leading to overfishing. Additionally, even if it is helpful to increase the number of edible aquatic animals, this could disrupt the ecological balance and not contribute to overall bio-“diversity" conservation.

The authors must clarify this in the revised version.

Reviewer #2: 1. The introduction is a little bit too long and may reduce readers’ attentions. I suggest simplifying this section and 4-5 paragraphs should be enough to describe the background and importance of the study, as well as objectives.

2. Honestly, I am not very familiar to methods of questionnaire surveys. I am wondering why there were only two paintings but not four assigned to each participant. It would be better to present the reason in the method.

3. Line 139: remove “,” following 166.

4. Result 3.2 (lines 339-342). More information should be present. For example, based on Table 2, both DIV and FSH appeared to be correlated with Self in most pictures. This findings were actually mentioned in the discussion section (line 425). In P3 and P4, ART showed a high correlation with ARS, especially for Exp and Cog, which was different from P1 and P2. These findings were also mentioned in the discussion section (lines 438-444) and it maybe interesting to explain this difference in the discussion.

5. Lines 375-384: this paragraph (the design and objectives of the study) would be better to present in introduction section than discussion section.

6. Actually, there were many different marine captures but only “fish” consumption were considered in this study. I know it is impossible to change the factor of fish consumption to the consumption of marine captures now, but it will be good to mention this limitation in the discussion section.

7. Subtitle: The case of paintings depicting aquatic biodiversity. Based on the paintings in this MS, it maybe not proper to use “aquatic biodiversity”. Words, like aquatic/marine animals, aquatic/marine captures, maybe better.

6. PLOS authors have the option to publish the peer review history of their article (what does this mean?). If published, this will include your full peer review and any attached files.

Reviewer #1: No

Reviewer #2: **Yes: **ZENG XIANYUAN

---

## [Author Response · Author response to Decision Letter 0]

24 Nov 2023

Responses to reviewers 

Reviewer #1: This manuscript explores the relationship between aesthetic experiences of art and experiences of nature, focusing on aquatic animals. The idea and perspective of this manuscript are intriguing, but some contents are not always clear. The authors need to clarify the following:

> We thank Reviewer 1 for his/her comments that helped us to improve the manuscript and clarify and the proposed conclusions. 

Specific comments:

1. Line 204-220. The use of Marks (1)-(7) for labeling may be confused with reference citation numbers. The way they are used in Line 307 and 308 appears to be clearer.

> We modified the labels accordingly (Lines 236-251). 

2. I suggest the author includes their questionnaire as supplementary information. This would greatly assist readers in quickly understanding their work.

> We added the questionnaire as supporting information (S2, translated from French to English). 

3. Based on their questionnaire, the authors concluded that “fish consumption and value attributed to fish as food had a significant positive impact one the aesthetic reception (Line 23-24, and also Line 293-293)”. Then they proposed that “taste and food consumption could be considered as a relevant nature aesthetic experience that elicits affective and emotional responses (Line 24-25)”. This conclusion and discussion seem reasonable.

However, on this basis, the authors emphasized that “taste and food consumption could be beneficial for biodiversity conservation (Line 26, and also Line 416-419)”. I found the logic behind this notion in the manuscript unclear. While it is true that these experiences would increase the attractivity of the paintings and decreases negative emotionality, it seems challenging to predict whether this would ultimately be beneficial or detrimental to biodiversity conservation. Someone might be more concerned about the sustainable conservation of biodiversity. Conversely, it may also arouse people's desires, potentially leading to overfishing. Additionally, even if it is helpful to increase the number of edible aquatic animals, this could disrupt the ecological balance and not contribute to overall bio-“diversity" conservation.

The authors must clarify this in the revised version.

> We agree with statement. The fact that aesthetic experience through taste is beneficial to conservation is not obvious, it is rather an idea or a perspective to be tested. For this reason, we deleted the mention of this hypothesis in the abstract (Line 32). In discussion, we put this statement into perspective by precising whether consumption might or might not be unfavorable to the motivation for the sustainable conservation of biodiversity (Lines 461-476). 

Reviewer #2: 

1. The introduction is a little bit too long and may reduce readers’ attentions. I suggest simplifying this section and 4-5 paragraphs should be enough to describe the background and importance of the study, as well as objectives.

> We thank Reviewer 2 for his/her comments that helped us to improve the manuscript regarding the introduction, methods, results and discussion. 

> We deleted the paragraph dedicated to neuro-aesthetic theories, since it was disconnected from the core subject of the study (Line 76). We also shortened the following paragraph (Line 97), and restructured the introduction in 4 paragraphs. 

2. Honestly, I am not very familiar to methods of questionnaire surveys. I am wondering why there were only two paintings but not four assigned to each participant. It would be better to present the reason in the method.

> We have specified the justification for this methodological choice in the “materials” section (Line 154-156): In order to limit the duration of the questionnaire and thus prevent the task from being too repetitive - and leading to biased responses -, the choice was made to assign only two paintings per participant instead of four.

3. Line 139: remove “,” following 166.

> Done

4. Result 3.2 (lines 339-342). More information should be present. 

- For example, based on Table 2, both DIV and FSH appeared to be correlated with Self in most pictures. These findings were actually mentioned in the discussion section (line 425). 

- In P3 and P4, ART showed a high correlation with ARS, especially for Exp and Cog, which was different from P1 and P2. These findings were also mentioned in the discussion section (lines 438-444) and it may be interesting to explain this difference in the discussion.

> We added comments related to Table 2 accordingly (Lines 382-388), mentioned again in discussion (Lines 486-488 and 504-510). 

5. Lines 375-384: this paragraph (the design and objectives of the study) would be better to present in introduction section than discussion section.

> This paragraph (Lines 421-430) aims to remind the objectives of the study before discussing the results in the details. If possible, we would like to keep this reminds here. 

6. Actually, there were many different marine captures but only “fish” consumption was considered in this study. I know it is impossible to change the factor of fish consumption to the consumption of marine captures now, but it will be good to mention this limitation in the discussion section.

> We added these limitations in conclusions (Lines 557-561). 

7. Subtitle: The case of paintings depicting aquatic biodiversity. Based on the paintings in this MS, it maybe not proper to use “aquatic biodiversity”. Words, like aquatic/marine animals, aquatic/marine captures, maybe better.

> We modified the title accordingly.

---

## [Editor Report · Decision Letter 1]

14 Dec 2023

PONE-D-23-28152R1

Nature experiences affect the aesthetic reception of art: The case of paintings depicting aquatic biodiversity

PLOS ONE

Dear Dr. Changeux,

Thank you very much for submitting your manuscript to PLOS ONE, and for responding to our recent requests regarding your submission. After careful evaluation, we have decided that your submission does not meet our publication criteria and must be rejected.

Based on our evaluation of your manuscript and the information you provided, we do not feel that your submission meets our ethical requirements for human subjects research submissions. PLOS ONE requires that research meets all applicable standards for the ethics of experimentation and research integrity (http://journals.plos.org/plosone/s/human-subjects-research). We reserve the right to reject any submission that does not meet our internal ethical standards, which in some cases are more stringent than local ethical standards.

You have not submitted the requested ethics approval documents. It is therefore not clear whether you obtained the necessary ethical approval for the study to take place. Please be aware that we expect all research involving human participants and/or medical data to have been approved by the authors' Institutional Review Board (IRB) or by an equivalent ethics committee(s).

As a result of these concerns we cannot consider the manuscript for publication at PLOS ONE. I am sorry that we cannot be more positive on this occasion, but hope that you understand the reasons for this decision.

Asmita Karmakar, PhD

- - - - -

---

## [Author Response · Author response to Decision Letter 1]

11 Mar 2024

A-Responses to reviewers (academics) 

Reviewer #1: This manuscript explores the relationship between aesthetic experiences of art and experiences of nature, focusing on aquatic animals. The idea and perspective of this manuscript are intriguing, but some contents are not always clear. The authors need to clarify the following:

> We thank Reviewer 1 for his/her comments that helped us to improve the manuscript and clarify and the proposed conclusions. 

Specific comments:

1. Line 204-220. The use of Marks (1)-(7) for labeling may be confused with reference citation numbers. The way they are used in Line 307 and 308 appears to be clearer.

> We modified the labels accordingly (Lines 236-251). 

2. I suggest the author includes their questionnaire as supplementary information. This would greatly assist readers in quickly understanding their work.

> We added the questionnaire as supporting information (S2, translated from French to English). 

3. Based on their questionnaire, the authors concluded that “fish consumption and value attributed to fish as food had a significant positive impact one the aesthetic reception (Line 23-24, and also Line 293-293)”. Then they proposed that “taste and food consumption could be considered as a relevant nature aesthetic experience that elicits affective and emotional responses (Line 24-25)”. This conclusion and discussion seem reasonable.

However, on this basis, the authors emphasized that “taste and food consumption could be beneficial for biodiversity conservation (Line 26, and also Line 416-419)”. I found the logic behind this notion in the manuscript unclear. While it is true that these experiences would increase the attractivity of the paintings and decreases negative emotionality, it seems challenging to predict whether this would ultimately be beneficial or detrimental to biodiversity conservation. Someone might be more concerned about the sustainable conservation of biodiversity. Conversely, it may also arouse people's desires, potentially leading to overfishing. Additionally, even if it is helpful to increase the number of edible aquatic animals, this could disrupt the ecological balance and not contribute to overall bio-“diversity" conservation.

The authors must clarify this in the revised version.

> We agree with statement. The fact that aesthetic experience through taste is beneficial to conservation is not obvious, it is rather an idea or a perspective to be tested. For this reason, we deleted the mention of this hypothesis in the abstract (Line 32). In discussion, we put this statement into perspective by precising whether consumption might or might not be unfavorable to the motivation for the sustainable conservation of biodiversity (Lines 461-476). 

Reviewer #2: 

1. The introduction is a little bit too long and may reduce readers’ attentions. I suggest simplifying this section and 4-5 paragraphs should be enough to describe the background and importance of the study, as well as objectives.

> We thank Reviewer 2 for his/her comments that helped us to improve the manuscript regarding the introduction, methods, results and discussion. 

> We deleted the paragraph dedicated to neuro-aesthetic theories, since it was disconnected from the core subject of the study (Line 76). We also shortened the following paragraph (Line 97), and restructured the introduction in 4 paragraphs. 

2. Honestly, I am not very familiar to methods of questionnaire surveys. I am wondering why there were only two paintings but not four assigned to each participant. It would be better to present the reason in the method.

> We have specified the justification for this methodological choice in the “materials” section (Line 154-156): In order to limit the duration of the questionnaire and thus prevent the task from being too repetitive - and leading to biased responses -, the choice was made to assign only two paintings per participant instead of four.

3. Line 139: remove “,” following 166.

> Done

4. Result 3.2 (lines 339-342). More information should be present. 

- For example, based on Table 2, both DIV and FSH appeared to be correlated with Self in most pictures. These findings were actually mentioned in the discussion section (line 425). 

- In P3 and P4, ART showed a high correlation with ARS, especially for Exp and Cog, which was different from P1 and P2. These findings were also mentioned in the discussion section (lines 438-444) and it may be interesting to explain this difference in the discussion.

> We added comments related to Table 2 accordingly (Lines 382-388), mentioned again in discussion (Lines 486-488 and 504-510). 

5. Lines 375-384: this paragraph (the design and objectives of the study) would be better to present in introduction section than discussion section.

> This paragraph (Lines 421-430) aims to remind the objectives of the study before discussing the results in the details. If possible, we would like to keep this reminds here. 

6. Actually, there were many different marine captures but only “fish” consumption was considered in this study. I know it is impossible to change the factor of fish consumption to the consumption of marine captures now, but it will be good to mention this limitation in the discussion section.

> We added these limitations in conclusions (Lines 557-561). 

7. Subtitle: The case of paintings depicting aquatic biodiversity. Based on the paintings in this MS, it maybe not proper to use “aquatic biodiversity”. Words, like aquatic/marine animals, aquatic/marine captures, maybe better.

> We modified the title accordingly. 

B- Responses to editorial decision

Subject: PLOS ONE Decision: PONE-D-23-28152R1 - [EMID:6507259d54a45917]

Dear Dr Karmakar,

Last December you have rejected our manuscript in reference because you were feeling that it didn’t comply with the ethical requirements for human subject research submissions of PLOS One.

After an in-depth examination of the issue you have raised, we suggest that you reconsider your position in the light of the new information below completed in response to the summary of requirements of your journal http://journals.plos.org/plosone/s/human-subjects-research)..

1-Obtain prior approval for human subjects research by an institutional review board (IRB) or equivalent ethics committee(s)

>>We can now provide you with the CNIL (French national commission for data processing and freedoms) declaration of conformity to the reference methodology framework MR-001, received on December 5, 2023, knowing that the ethic committee of Aix-Marseille University has adopted the following (translation from https://www.univ-amu.fr/fr/public/comite-dethique). 

“2-OUTSIDE THE JARDE LAW AND THE ETHICS COMMITTEE'S JURISDICTION

Research involving the human person within the meaning of II of article R1121-1 of the Public Health Code (...) - opinion surveys, pure observations in the social, ethnological or ethological sciences.”

Our on line questionnaire study is clearly an opinion survey and do not enter ethic committee jurisdiction.

2-Submit documentation from the review board or ethics committee confirming approval of the research. Identifying information about study participants must be redacted from the approval document before it is submitted to the journal”

>> We do not have any information enabling to identify the participants, as the online questionnaire was anonymous.

3-Declare compliance with ethical practices upon submission of a manuscript

>>You will find in the end of this letter a copy of PLOS One Human Participants Research Checklist we have signed and kept at your disposal.

4-Report details on how informed consent for the research was obtained (or explain why consent was not obtained)

>> Participants gave their informed consent by checking a box at the beginning of the questionnaire stating: “Knowing the information transmitted to me, I freely and voluntarily agree to participate in the research project entitled: “Aesthetic reception of aquatic biodiversity in art.””

5-For clinical trials, provide trial registration details, the study protocol, and CONSORT documentation (more information below)

>> It is not a clinical trial.

6-Confirm that an identified individual has provided written consent for the use of that information

>> We confirm that, even if they are not identifiable, the participants have given their consent by ticking the box as indicated in previous §4.

Our manuscript had already been the subject of a previous review (PONE-D-23-28152R1, to which we responded to each point raised by the reviewers), but following your comments, we have modified it slightly (line 205-207).

If you agree to reconsider your decision, we offer you the opportunity to submit a new version and the original .pdf documents mentioned in this letter.

Sincerely,

Thomas CHANGEUX, Ph D

Copy of PLOS One One Human Participants Research Checklist

Complete the following if your study involved human participants or human participants’ data. These questions should be addressed for prospective and retrospective studies.

1.Did you obtain ethics approval for this study?

•If yes, please upload (file type “Other”) the original approval document you received from your ethics committee. If the original document is in another language, please also provide an English translation.

>>English translation of the document: 

“CNIL (national commission for data processing and freedoms) declaration of conformity 

Declaration of conformity to the reference methodology framework MR-001, received on December 5, 2023

By this declaration, the declarant certifies the conformity of his/her processing(s) of personal data with the reference system mentioned above. The CNIL may at any time verify, by mail or by means of an on-site or online inspection, the conformity of this processing(s).”

X Uploaded 

2. If you prospectively recruited human participants for the study – for example, you conducted a clinical trial, distributed questionnaires, or obtained tissues, data or samples for the purposes of this study, please report in the Methods:

i. the day, month and year of the start and end of the recruitment period for this study.

>> The study was carried out from September 2022 to November 2022 (see L. 205)

ii.whether participants provided informed consent, and if so, what type was obtained (for instance, written or verbal, and if verbal, how it was documented and witnessed). If your study included minors, state whether you obtained consent from parents or guardians. If the need for consent was waived by the ethics committee, please include this information.

>> The information is now included L. 205 – 207 ou our new manuscript: “The study was carried out on an adult audience (over 18 years old), recruited by an email information campaign among Aix-Marseille University staff and national academic mailing lists. Participants gave their informed consent by checking a box at the beginning of the questionnaire stating: Knowing the information transmitted to me, I freely and voluntarily agree to participate in the research project entitled: “Aesthetic reception of aquatic biodiversity in art”.”

_X__ Completed ___ N/A

3.If you are reporting a retrospective study of medical records or archived samples, please report in the Methods section:

i. the day, month and year when the data were accessed for research purposes

ii. whether authors had access to information that could identify individual participants during or after data collection

___ Completed _X__ N/A: not concerned

Signed Thomas CHANGEUX

---

## [Editor Report · Decision Letter 2]

22 Apr 2024

Nature experiences affects the aesthetic reception of art:  The case of paintings depicting aquatic animals

PONE-D-23-28152R2

Dear Dr. Changeux,

We’re pleased to inform you that your manuscript has been judged scientifically suitable for publication and will be formally accepted for publication once it meets all outstanding technical requirements.

Kind regards,

Avanti Dey, PhD

Staff Editor

PLOS ONE

Additional Editor Comments (optional):

Its an ethical issue that has ben raised very relevant by the Editorial team and now I can see that you have tried best to addressed the issue. However it s needs to be confirmed once again whether its is violating any further terms and condition of the journal.
---

## [Editor Report · Acceptance letter]

9 Jul 2024

PONE-D-23-28152R2 

PLOS ONE

Dear Dr. Changeux, 

I'm pleased to inform you that your manuscript has been deemed suitable for publication in PLOS ONE. Congratulations! Your manuscript is now being handed over to our production team.

Kind regards, 

on behalf of

Dr. Avanti Dey 

Staff Editor

PLOS ONE